# Land Use and Land Cover Change Determinants in Raya Valley, Tigray, Northern Ethiopian Highlands

Eskinder Gidey [1], Oagile Dikinya [2], Reuben Sebego [2], Eagilwe Segosebe [2], Amanuel Zenebe [1], Said Mussa [3], Paidamwoyo Mhangara [4] and Emiru Birhane [1,5,*]

[1]  Department of Land Resources Management and Environmental Protection, College of Dryland Agriculture and Natural Resources, Mekelle University, Mekelle P.O. Box 231, Ethiopia
[2]  Department of Environmental Science, Faculty of Science, University of Botswana, Private Bag, Gaborone 00704, Botswana
[3]  Department of Statistics, College of Natural and Computational Sciences, Mekelle University, Mekelle P.O. Box 231, Ethiopia
[4]  School of Geography, Archaeology and Environmental Studies, Faculty of Science, Witwatersrand University, Johannesburg, Private Bag 3, Wits 2050, South Africa
[5]  Institute of Climate and Society, Mekelle University, Mekelle P.O. Box 231, Ethiopia
*  Correspondence: emiru.birhane@mu.edu.et; Tel.: +251–914702336

**Abstract:** Land use and land cover change (LULCC) is the result of both natural and socio-economic determinants. The aim of this study was to model the determinant factors of land cover changes in Raya Valley, Southern Tigray, Ethiopia. Multistage sampling was used to collect data from 246 households sampled from lowlands (47), midlands (104), highlands (93), and sub-alpine (2) agro-climatological zone. Descriptive statistics and logit regression model were used to analyze the field survey data. Agricultural land expansion, fuelwood extraction, deforestation, overgrazing and expansion of infrastructure were the proximate causes of LULCC in the study area. Agricultural land expansion ($p = 0.084$) and wood extraction for fuel and charcoal production ($p = 0.01$) were the prominent causes for LULCC. Persistent drought ($p = 0.001$), rapid population growth ($p = 0.027$), and climate variability ($p = 0.013$) were the underlying driving factors of LULCC. The determinants of LULCC need to be considered and mitigated to draw robust land use policy for sustainable land management by the smallholder farmers. This study provides important results for designing and implementing scientific land management strategies by policy makers and land managers.

**Keywords:** agricultural land expansion; deforestation; LULCC drivers; logit model; Raya Valley; Southern Tigray

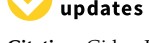



## 1. Introduction

Land use and land cover change (LULCC) is the result of both natural and socio-economic factors [1,2]. These factors have changed the nature of the environment at an alarming rate due to human activities [3]. Therefore, LULCC has remained as one of the global and regional environmental challenges that have caused social, economic, and political crises. The dynamics of LULCC have not been identical in all parts of the globe due to several driving factors [4], and they have become sustainable research topics [5]. This explains why LULCC processes are becoming very complex, with causes and effects operating at different temporal and spatial scales [6,7]. It remains a key research challenge throughout the world in general and in developing countries in particular. The determinants of LULCC are not well understood mainly in human–nature relations [8]. There is a need to identify the proximate cause and underlying driving factors of LULCC, which are driven by interrelated factors, including national policies [9,10]. Population growth, expansion of farmland, inappropriate land management, civil war, and fuelwood demand are some of the major triggering factors of LULCC leading to land degradation (e.g., soil

erosion, desertification, loss of biodiversity) and food insecurity [11]. However, the causes of LULCC vary from location to location, depending on the location-specific factors [12]. Thus, the LULCC could not be accounted for by a single factor [13,14]. For example, in 4% of cases in Asia, a single factor (e.g., agricultural expansion) due to high demand for cultivation explained the cause of LULCC in 30% of cases, two factors (mainly agricultural expansion and wood extraction) were identified, while in 45% of cases, three factors related to deforestation were identified [15]. The observed LULCC was driven mainly by the combination of both proximate causes and underlying driving factors. Proximate causes are explained by the immediate actions imposed by humans at the local level (e.g., the direct impacts of agricultural land expansion on forest cover), while the underlying driving forces are largely defined as the fundamental social processes (e.g., human population dynamics) operating at the local level [16]. The underlying driving forces have indirect impact [16] and accelerate the impacts of proximate causes. In addition, the proximate cause of LULCC is defined as a factor, which constitutes a direct cause of the phenomenon to be explained, and the underlying driving factor is also defined as a factor, which causes the proximate causes of LULCC [17]. Both of the factors are caused due to human interaction with the environment.

In Africa, few studies on LULCC driving forces have been conducted for sustainable use of land resources, and these studies are useful to investigate the implication of LULCC in several sectors (e.g., agriculture) [18,19]. One of the major implications of LULCC in Africa is accelerated land degradation phenomena. Therefore, identification of LULCC factors using an integrated approach may help to ensure better land use planning and environmental sustainability. For instance, Ethiopia is currently facing severe land scarcity, particularly in the northern highlands area due to high population growth [20,21], increased population density, smaller farm sizes, and settlement expansion, resulting in persistent LULCC. Nearly 36.3% of agricultural land, consisting of arable land (15.2%), permanent crops such as coffee, rubber, citrus (1.1%), permanent pasture (20%) and others (51.5%), is the key economic stay in the country [22]. Currently, more than 83% of Ethiopians derive their livelihoods directly from land resources, and with a population growth rate of almost 2.7% per year, food production is also expected to increase at least at the same rate [23]. For instance, [24] reported that lack of effective land use policy in Ethiopia has been significantly affected by the cultivation and productivity of the land. The same authors have also reported that the natural forest cover decreased from 27% in 1957 to 2% in 1982 and to 0.3% in 1995. Studies reported that the forest coverage of Ethiopia has been declining at a rate of 0.8% from 1990 to 2015 due to LULCC [25]. All of these factors have a significant impact on the economic condition of the country. However, the scale of the causes and consequences of LULCC varied spatially and were not uniform in all parts of Ethiopia [4,18]. At this time, a comprehensive study that integrates both natural and anthropogenic factors of LULCC is limited [26] mainly in Ethiopia. For instance, [22] reported that significant land use and land cover change has been observed in Raya Valley, Southern Tigray, from 1984 to 2015. Both grassland and water bodies have been also declining by 36.9% and 8.7%, respectively [22]. This decline may have an effect on the smallholder farmers in the study area because they are losing their potential agricultural production. Traditional agricultural production cannot meet the needs of the people's daily life as a result; it entails the development of science and technology to maximize agricultural productivity [3]. The driving forces of LULCC were studied in the Yanhe River Basin, China, from three points of view: (i) population and urbanization, (ii) regional economic development, and (iii) ecological restoration and governance policies [3]. The socio-economic driving forces of land use change in Kunshan, Yangtze River Delta economic area of China, was investigated [19]. Moreover, the trends of LULCC and their driving forces in the Kilombero valley floodplain, southeastern Tanzania, were examined [27]. Furthermore, analyses of the trends and drivers of LULCC in western Ethiopia using 15 focus group discussions and 32 key informants guided by checklists were conducted [28]. A study on LULCC and its driving forces in the Shenkolla watershed, south central Ethiopia

was carried out [7]. However, none of the studies were supported with robust statistical models (e.g., logit) to determine the proximate and underlying driving forces of LULCC, but the discussants were trying to identify the factors mainly in western Ethiopia. The logit model is simple and more efficient in identifying and predicting the socio-economic and biophysical driving forces of LULCC.

Currently, policymakers are seeking research–based information on the root causes of LULCC in order to develop scientific remedial actions [29]. The Ethiopian Rift Valley, which covers portions of the study area, is experiencing land use and land cover change, drought and high population pressure [30]. The LULCC is therefore the result of complex interactions between a variety of driving factors between human activity and biophysical factors [31,32]. This change affects the biophysical environment due to changes in land use but is determined by socio-economic driving forces [33]. Socio-economic information including population size and density, land size, and education are some of the driving forces of LULCC. A closer understanding of the existing biophysical factors including climate, elevation, aspect, slope, and soil type is required to characterize the perceptions of local people in LULCC driving forces [30,33,34].

Understanding the driving force of LULCC at a local level is essential to grasp the comprehensive and reliable information for the sustainable use of land resources [23,28]. Conversely, detailed socio-economic data supported with remote sensing products have the capability to improve the investigation of both proximate causes and underlying driving factors of LULCC. It is useful to distinguish the complex set of socio-economic and biophysical forces that influence the rate and spatial pattern of land use change [35]. In addition, a detailed understanding of the drivers of LULCC and their interlinkages improves intervention and avoids a decline in natural resources [8,12]. The aim of this study was to determine both the proximate causes and underlying driving factors of the LULCC perceived by smallholder farmers in Raya Valley, Tigray, Ethiopia. The finding of this study helps to provide better insight for decision makers and land use planners to implement suitable land management policies and strategies based on the trends of LULCC in the study area. It is also helpful to the local communities of the study area to diminish the impacts of LULCC and to control land degradation processes.

## 2. Methods

### 2.1. Study Site

This research was conducted in Raya Valley, Southern Tigray, Ethiopia. Geographically, it is situated at 39°0′0″ and 40°52′30″ longitude easting and 12°7′30″ and 13°12′0″ latitude northing (Figure 1). The total land mass of the research area is about 14,532 km$^2$. The altitude ranges from 324 to 4129 m above sea level (m.a.s.l). Rainfall is erratic and bimodal [36]. The mean annual rainfall of the research site reaches up to 558 mm [22]. Furthermore, the maximum and minimum temperatures were between 30.5 and 15.9 °C in 2015 [22]. The study area has various land cover types. Cultivated land and shrub/bushlands are the main land cover types that cover approximately 6232.3 (42.9%) and 3547.3 km$^2$ (24.4%), while others cover 4752.5 km$^2$ (32.7%). The total population of the study area was 1,200,136 (CSA, 2007) with 604,063 (50.3%) male and 596,073 (49.7%) female. The maximum and minimum family sizes of the study area are 6.4 and 4.2, respectively (Table 1). In addition, the total household of the study area is 272,295 for both male-headed and female-headed households (Table 1). The overall annual population growth rate during the periods of 1994 and 2007 was 2.6% per year [37]. After 2007, the population grew at an annual rate of 2.7%. Moreover, agriculture is the key economic activity in the area. The small-scale farmers in the area practice mixed farming systems such as crop production and animal rearing as a main source of livelihood. The livelihood zone in the area is largely classified as agro-pastoral, pastoral and cropping. According to the Raya Valley Livelihood Zone (2007), the dominant crops produced in the study area are sorghum, teff, and maize. Of all crops, sorghum is widely used as a staple food and is broadly grown under rain-fed agricultural practices, while teff is produced for both food and cash crop [22]. Cattle, goats,

and sheep are extensively reared as a source of revenue and food. The farming activity largely depends on Belg (small rain) and Kiremt (Meher), the main rainy season.

**Figure 1.** Location of the study area showing the different land features and elevation.

**Table 1.** Total households and family size of the study area by districts.

| S. No. | Site | Households | | | | Total Household | Family Size |
|---|---|---|---|---|---|---|---|
| | | Urban | % | Rural | % | | |
| 1. | Megale | 209 | 0.8 | 4475 | 1.8 | 4684 | 6.0 |
| 2. | Yalo | 194 | 0.8 | 7911 | 3.2 | 8105 | 5.9 |
| 3. | Gulina | 831 | 3.3 | 6989 | 2.8 | 7820 | 6.4 |
| 4. | Gidan | 2420 | 9.5 | 34,889 | 14.1 | 37,309 | 4.2 |
| 5. | Kobo | 9398 | 36.8 | 44,841 | 18.2 | 54,239 | 4.1 |
| 6. | Alaje | 2118 | 8.3 | 22,629 | 9.2 | 24,747 | 4.4 |
| 7. | Alamata | 1283 | 5.0 | 19,212 | 7.8 | 20,495 | 4.2 |
| 8. | Hintalo–Wejirat | 3411 | 13.4 | 30,868 | 12.5 | 34,279 | 4.5 |
| 9. | Ofla | – | 0.0 | 29,525 | 12.0 | 29,525 | 4.3 |
| 10. | Endamehoni | 904 | 3.5 | 17,894 | 7.3 | 18,798 | 4.5 |
| 11. | Raya Azebo | 4739 | 18.6 | 27,555 | 11.2 | 32,294 | 4.2 |
| | Total | 25,507 | 100 | 246,788 | 100 | 272,295 | - |

### 2.2. Sampling Procedure

A multi-stage sampling technique was used to determine the appropriate sampling size. A purposive sampling was first employed to examine the multidimensional problems that the rural households of Raya Valley and its environments are facing. The majority of households reside in lowland, midland, highlands and sub-alpine agro-climatological zones, and they have similar cultures and sources of income. Agriculture is the main source

of income. A cluster sampling approach based on the homogeneity of the area was used to determine the number of households in similar agro-climatological zones. According to the Ethiopian Ministry of Agriculture (MoA) agro-climatological zones (Table 2), an agro-climatological sampling technique based on stratification was used. Using Ref. [38], a sampling technique and size determination equation were then used to sample 20% of households (Equation (1)). A systematic random sampling method based on the homogeneity of the area was then applied to proportionally sample a total sample of 246 household respondents (Table 2).

$$n = \frac{z^2 (p)(q)}{(d^2)} \tag{1}$$

where $n$ = desired sample size, $z$ = confidence level, 95%, $p$ = proportion of households (20%), $q$ = 1–0.2 i.e., 0.80, $d$ = acceptable error 5%.

**Table 2.** Sample size and area coverage distributed across agro-climatology in Raya Valley, Southern Tigray, Ethiopia.

| Altitude (m) | Agro-Ecology | Household (Number) | Household (%) | Sample Size (Number) | Sample Size (%) | Area (km$^{2)}$) | Area (%) |
|---|---|---|---|---|---|---|---|
| <500 | Desert | 0.0 | 0.0 | 0.0 | 0.0 | 233.5 | 1.6 |
| 500–1500 | Lowlands | 52,236.0 | 19.2 | 47.0 | 19.1 | 5416.8 | 37.3 |
| 1500–2300 | Midlands | 114,869.0 | 42.2 | 104.0 | 42.3 | 5428.9 | 37.4 |
| 2300–3200 | Highlands | 103,324.0 | 37.9 | 93.0 | 37.8 | 3051.0 | 21.0 |
| 3200–3700 | Sub–alpine | 1679.0 | 0.6 | 2.0 | 0.8 | 372.5 | 2.6 |
| >3700 | Alpine | 187.0 | 0.1 | 0.0 | 0.0 | 29.3 | 0.2 |
| Total | | 272,295 | 100 | 246 | 100 | 14,532.0 | 100 |

### 2.3. Field Survey Data Acquisition

The field survey data were collected using a semi-structured questionnaire, interviews with key informants such as government officials, focus group discussions with local residents (elders), and field observation. The data were used to collect comprehensive information on how the farmers use their lands, if there has been any change in their land use system, and why it is changing from time to time. This field survey was used to assess the ideas, beliefs, and/or opinions of the farmers on the proximate and underlying driving forces of LULCC. Ref. [39] reported that the LULCC theory is needed to conceptualize the relationship between driving forces, the mitigation process, and human behavior. As a result, a small group of five to ten farmers supported by the discussion guideline and a moderator were identified to conduct a focus group discussion in the study area. Each interview took about 55 min. In addition, demographic data were collected from the Central Statistical Agency of Ethiopia for the period 1994(5), 2007 and 2015 to quantify the population density.

### 2.4. Land Use and Land Cover Change Data

The LULCC data of Raya Valley were obtained from [22] for the period 1984, 1995 and 2015 to determine the proximate causes and underlying driving forces of LULCC in the area (Figure 2). The descriptions of each land use and land cover are presented in Table 3. The data indicated that there were significant changes in most of the land use and land cover types during the last three decades (Table 4). An increased annual change rate trend was observed in shrub/bush lands by 150.3 km$^2$ (5.33%), grasslands 21.6 km$^2$ (6.59%), built up area 8.8 km$^2$ (9.19%), forestland 6.6 km$^2$ (5.73%), barren land 2.5 km$^2$ (0.15%), and water body 0.6 km$^2$ (1.19%). However, the croplands and floodplain were decreased annually by 118.3 (1.95%) and 72.1 km$^2$ (2.47%), respectively. Furthermore, during the period 1995–2015, the annual change rate of the shrub/bushlands was 46.4 km$^2$ (1.0%), water body 0.6 km$^2$ (1.0%), grasslands 17.9 km$^2$ (3.2%), and floodplain 68.8 km$^2$ (3.2%). Conversely, the croplands, forestlands, built–up areas, and barren lands were increased

annually by 72.9 (1.5%), 2.8 (1.5%), 20 (10.4%), and 37.9 km$^2$ (1.8%), respectively. Moreover, during the year 1984–2015, increases in croplands by 5 km$^2$ (0.08%), forestland by 4.2 km$^2$ (3.62%), shrub/bushland 23.4 km$^2$ (0.83%), built–up area 16 km$^2$ (16.76%), and barren lands 25.3 km$^2$ (1.19%) were observed. In the same year, water bodies, grasslands, and floodplain area were declined annually by 0.1 (0.28%), 3.9 (1.19%), and 69.9 km$^2$ (2.4%), respectively (Figure 2 and Table 4).

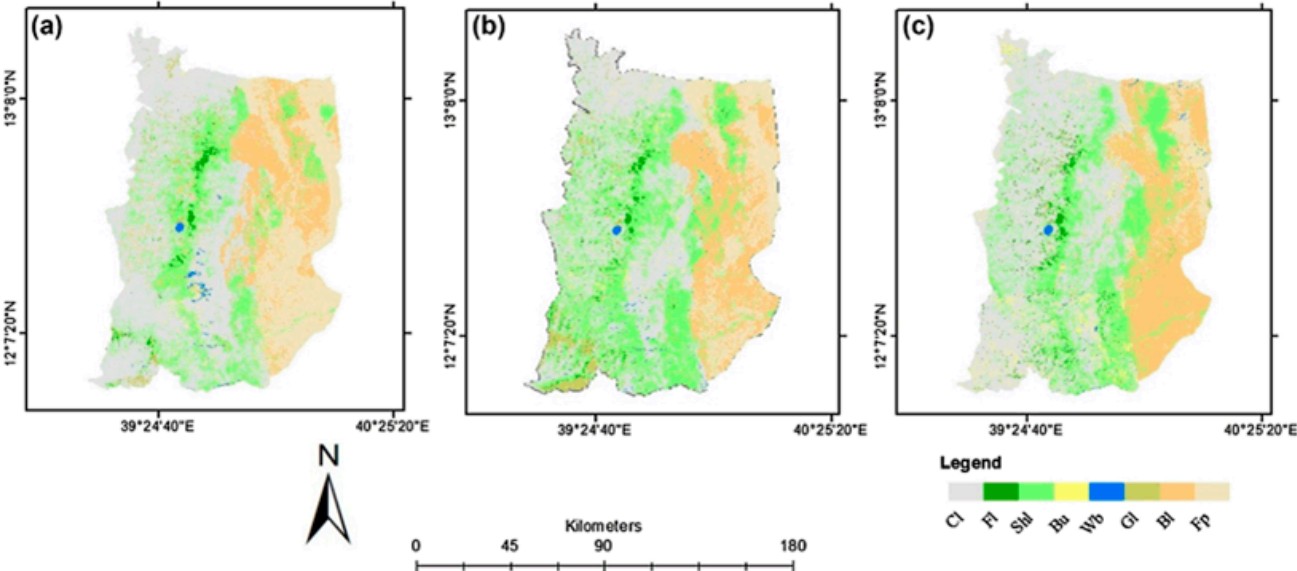

**Figure 2.** Land use and land cover types of the study area in 1984 (**a**), 1995 (**b**), and 2015 (**c**) [22].

**Table 3.** Description of land use and land cover types in the study area [22].

| Land Use and Land Cover Types | Description |
|---|---|
| Cropland (Cl) | Those regularly used to grow domesticated plants, ranging from the long-fallow, land-rotational systems to permanent, intensively, moderately, and sparsely cultivated land |
| Forest land (Fl) | Land spanning more than 0.5 hectares with trees higher than 5 m and a canopy cover of more than 10 percent, or trees able to reach these thresholds in situ. |
| Shrub/bushland (Shl/bl) | Woody perennial plant, generally more than 0.5 m and less 5 m in height at maturity and without a definite crown. Areas with a cover of shrubs and short trees mixed with grasses |
| Built up area (Bu) | Residential, urban area, commercial, and industrial |
| Water body (Wb) | Inland water bodies generally include major rivers, lakes and water reservoirs |
| Grassland (Gl) | Land with herbaceous types of cover; tree and shrub cover is less than 10% |
| Barren land (Bl) | Land with exposed soil, sand, rocks, or snow and never have more than 10% vegetated cover at any time of the year |
| Floodplain (Fp) | Flat area of land next to a river or stream covered by the lower course of the river, carrying a large volume of water during the rainy season, but covered most of the year by sand and different sizes of gravel and stones |

**Table 4.** Land use and land cover change of the study area [22].

| LC | 1984 | | 1995 | | LULCC from 1984–1995 | | | |
|---|---|---|---|---|---|---|---|---|
| | Area in km² | % | Area in km² | % | Area in km² | % | Annual Change Rate in km² | % |
| Cl | 6076 | 41.8 | 4774.8 | 32.9 | −1301.2 | −21.4 | −118.3 | −1.95 |
| Fl | 115.1 | 0.8 | 187.6 | 1.3 | +72.5 | 63 | +6.6 | +5.73 |
| Shl/bl | 2821.9 | 19.4 | 4474.8 | 30.8 | +1652.9 | 58.6 | +150.3 | +5.33 |
| Bu | 95.7 | 0.7 | 192.4 | 1.3 | +96.8 | 101.1 | +8.8 | +9.19 |
| Wb | 50.6 | 0.3 | 57.3 | 0.4 | +6.6 | 13.1 | +0.6 | +1.19 |
| Gl | 328.3 | 2.3 | 566.1 | 3.9 | +237.8 | 72.4 | +21.6 | +6.59 |
| Bl | 2129.1 | 14.7 | 2156.3 | 14.8 | +27.2 | 1.3 | +2.5 | +0.12 |
| Fp | 2915.6 | 20.1 | 2122.9 | 14.6 | −792.7 | −27.2 | −72.1 | −2.47 |
| *Total* | *14,532* | *100* | *14,532* | *100* | - | - | - | - |

| LC | 1995 | | 2015 | | LULCC from 1995–2015 | | | |
|---|---|---|---|---|---|---|---|---|
| | Area in km² | % | Area in km² | % | Area in km² | % | Annual Change Rate in km² | % |
| Cl | 4774.8 | 32.9 | 6232.3 | 42.9 | +1457.5 | +30.5 | +72.9 | +1.5 |
| Fl | 187.6 | 1.3 | 244.2 | 1.7 | +56.6 | +30.2 | +2.8 | +1.5 |
| Shl/bl | 4474.8 | 30.8 | 3547.3 | 24.4 | −927.4 | −20.7 | −46.4 | −1.0 |
| Bu | 192.4 | 1.3 | 592.9 | 4.1 | +400.4 | +208.1 | +20 | +10.4 |
| Wb | 57.3 | 0.4 | 46.2 | 0.3 | −11.0 | −19.3 | −0.6 | −1.0 |
| Gl | 566.1 | 3.9 | 207.2 | 1.4 | −359.0 | −63.4 | −17.9 | −3.2 |
| Bl | 2156.3 | 14.8 | 2914.8 | 20.1 | +758.5 | +35.2 | +37.9 | +1.8 |
| Fp | 2122.9 | 14.6 | 747.2 | 5.1 | −1375.7 | −64.8 | −68.8 | −3.2 |
| *Total* | *14,532* | *100* | *14,532* | *100* | - | - | - | - |

| LC | 1984 | | 2015 | | LULCC from 1984–2015 | | | |
|---|---|---|---|---|---|---|---|---|
| | Area in km² | % | Area in km² | % | Area in km² | % | Annual Change Rate in km² | % |
| Cl | 6076 | 41.8 | 6232.3 | 42.9 | 156.3 | +2.6 | +5.0 | +0.08 |
| Fl | 115.1 | 0.8 | 244.2 | 1.7 | 129.1 | +112.2 | +4.2 | +3.62 |
| Shl/bl | 2821.9 | 19.4 | 3547.3 | 24.4 | 725.5 | +25.7 | +23.4 | +0.83 |
| Bu | 95.7 | 0.7 | 592.9 | 4.1 | 497.2 | +519.7 | +16.0 | +16.76 |
| Wb | 50.6 | 0.3 | 46.2 | 0.3 | −4.4 | −8.7 | −0.1 | −0.28 |
| Gl | 328.3 | 2.3 | 207.2 | 1.4 | −121.1 | −36.9 | −3.9 | −1.19 |
| Bl | 2129.1 | 14.7 | 2914.8 | 20.1 | 785.7 | +36.9 | +25.3 | +1.19 |
| Fp | 2915.6 | 20.1 | 747.2 | 5.1 | −2168.4 | −74.4 | −69.9 | −2.40 |
| *Total* | *14,532* | *100* | *14,532* | *100* | - | - | - | - |

### 2.5. Data Analysis

The field survey data were analyzed in STATA v.14 once it was cleaned and coded. Both descriptive statistics such as chi-square, mean, standard deviation, maximum, percentage and logit model were used to analyze the field survey data to determine the main LULCC driving factors in the study area (Equation (2)).

$$Logit(Y) = \alpha + \beta_1 X_1 + \beta_2 X_2 + \beta_3 X_3 \ldots \ldots + \beta_n X_n \tag{2}$$

where $Y$ = LULCC (response variable), $\alpha$ = intercept, $\beta_1 \ldots \beta_n$ = coefficient of each factor variable, $X_1 \ldots X_n$ = predictors variable.

The land use and land cover change (LULCC) ($Y$) was considered as a response variable because the change occurred due to the influence of several factors. These factors or predictors were taken as an independent variable ($X$), such as expansion of agricultural land, extraction of fuelwood, drought, population growth, climate change, topography, unemployment, lack of land use policies, overgrazing, deforestation, and expansion of infrastructure.

## 3. Results and Discussion

### 3.1. Household Characteristics

Characterizations of household heads by age group and gender have paramount significance to comprehending respondents' views on the driving factors of land use and land cover change at different levels. The maximum, minimum, and mean ages of respondents were 74, 41, and 54, respectively. The results indicated that nearly 204 (82.9%) of the respondents were male–headed household, while the female-headed households were about 42 (17.1%). In both sexes, about 218 (88.62%) of the respondents were older adults (45–65 years old) and elderly (>65 years old). This provides a great opportunity to conduct an in-depth study on LULCC, as the respondents can recall the change in the land use system. In the study area, about 60.98% (150) of the respondents were illiterate, while 39.02% (96) of respondents were literate. Literate households can easily understand the causes of land use and land cover changes driving factors. The maximum and minimum land holding sizes of the household respondents were between 0 and 1 hectare. However, the mean land size owned by farmers was 0.54 ha, and this land provides poor crop production due to the poor soil productivity or the fertility of the land, lack of moisture, erratic rainfall, and high temperature, among others. This all causes acute food shortages in each household because of the large family size in the region. The average family size per household was 4.4.

### 3.2. Agro-Climatological-Based Farmers' Perceptions on the Determinants of LULCC

All the respondents residing in the lowlands, midlands and highlands had a similar understanding toward the driving forces of LULCC of their community. Respondents from the lowlands had described more driving force than the mid- and highlands. Settlement expansion, climate change and variability, high population pressure, land degradation, recurrent drought, infrastructure expansion, mining, deforestation, overgrazing, lack of effective land use policy, agricultural land intensification, topography, and unemployment were identified as the major driving factors across all agro-climatological zones in the study area (Table 5). Both fuelwood extraction and overgrazing were the possible determinant factors of LULCC followed by the lack of land use policy and agricultural land intensification (Table 5). The collection of fuelwood for domestic use was noted as one of the causes of deforestation in Africa [16]. Drought was also mentioned as another notable determinant factor, while urbanization and stone quarries were the potential factors of LULCC in the highlands. Moreover, the associations among the various possible determinant factors such as agricultural land intensification, climate change/variability, across the three agro-climatological zones were tested using chi–square test of significance. The results presented in Table 5 show that there was statistically significant association among the various determinant factors such as agricultural land intensification ($\chi^2 = 4.9, df = 2, p - \text{value} = 0.086$), urbanization $\chi^2 = 9.9, df = 2, p - \text{value} = 0.008$), government land use policy ($\chi^2 = 8.18, df = 2, p - \text{value} = 0.017$), fuelwood extraction ($\chi^2 = 5.36, df = 2, p - \text{value} = 0.069$), overgrazing ($\chi^2 = 15.09, df = 2, p - \text{value} = 0.001$), drought ($\chi^2 = 22.23, df = 2, p - \text{value} = 0.0001$), and stone quarry ($\chi^2 = 10.99, df = 2, p - \text{value} = 0.004$).

**Table 5.** Response of households on determinants of LULCC across agro-climatology in Raya Valley, Southern Tigray, Ethiopia.

| Determinant Factors | Household Response | Agro-Climatological Zone | | | | | | Chi-Square | *p*-Value |
|---|---|---|---|---|---|---|---|---|---|
| | | Lowland | | Midland | | Highland | | | |
| | | Freq. | % | Freq. | % | Freq. | % | | |
| Agricultural land intensification | No | 138 | 80.2 | 27 | 64.3 | 25 | 78.1 | 4.899 | 0.086 |
| | Yes | 34 | 19.8 | 15 | 35.7 | 7 | 21.9 | | |
| Climate Change/Variability | No | 2 | 1.2 | 1 | 2.4 | 0 | 0.0 | 0.870 | 0.647 |
| | Yes | 170 | 98.8 | 41 | 97.6 | 32 | 100.0 | | |
| Settlement expansion | No | 23 | 13.4 | 4 | 9.5 | 1 | 3.1 | 2.982 | 0.225 a |
| | Yes | 149 | 86.6 | 38 | 90.5 | 31 | 96.9 | | |
| Urbanization | No | 110 | 64.0 | 25 | 59.5 | 11 | 34.4 | 9.785 | 0.008 * |
| | Yes | 62 | 36.0 | 17 | 40.5 | 21 | 65.6 | | |
| Government Land use policy | No | 46 | 26.7 | 3.0 | 7.1 | 10 | 31.3 | 8.179 | 0.017 * |
| | Yes | 126 | 73.3 | 39 | 92.9 | 22 | 68.8 | | |
| Population growth/pressure | No | 6.0 | 3.5 | 1.0 | 2.4 | 0.0 | 0.0 | 1.227 | 0.541 a,b |
| | Yes | 166 | 96.5 | 41 | 97.6 | 32 | 100.0 | | |
| Land degradation | No | 15 | 8.7 | 4 | 9.5 | 1 | 3.1 | 1.263 | 0.532 |
| | Yes | 157 | 91.3 | 38 | 90.5 | 31 | 96.9 | | |
| Deforestation | No | 10 | 5.8 | 2 | 4.8 | 1 | 3.1 | 0.417 | 0.812 a |
| | Yes | 162 | 94.2 | 40 | 95.2 | 31 | 96.9 | | |
| Fuelwood extraction | No | 16 | 9.3 | 0 | 0.0 | 1 | 3.1 | 5.360 | 0.069 a |
| | Yes | 156 | 90.7 | 42 | 100.0 | 31 | 96.9 | | |
| Overgrazing | No | 35 | 20.3 | 0 | 0.0 | 1 | 3.1 | 15.089 | 0.001 * |
| | Yes | 137 | 79.7 | 42 | 100.0 | 31 | 96.9 | | |
| Drought | No | 4 | 2.3 | 8 | 19.0 | 0 | 0.0 | 22.229 | 0.000 a,* |
| | Yes | 168 | 97.7 | 34 | 81.0 | 32 | 100.0 | | |
| Lack of employment | No | 55 | 32.0 | 10 | 23.8 | 7 | 21.9 | 2.059 | 0.357 |
| | Yes | 117 | 68.0 | 32 | 76.2 | 25 | 78.1 | | |
| Infrastructure expansion | No | 66 | 38.4 | 17 | 40.5 | 15 | 46.9 | 0.823 | 0.663 |
| | Yes | 106 | 61.6 | 25 | 59.5 | 17 | 53.1 | | |
| Stone quarry (mining) | No | 140 | 81.4 | 41 | 97.6 | 22 | 68.8 | 10.996 | 0.004 * |
| | Yes | 32 | 18.6 | 1 | 2.4 | 10 | 31.3 | | |

Note: a,b and * represent the statistical significance levels at 5%.

### 3.3. Proximate Causes

The direct human activities of LULCC at the local level include agricultural expansion, which originates from the planned land use and directly affects the forest cover [40,41]. Understanding of the proximate causes thus helps to predict the future LULCC [42]. Agricultural land expansion, fuelwood extraction, and infrastructure extension are major proximate causes of land use and land cover changes [40]. However, in this study, agricultural expansion, wood extraction for fuel and charcoal production, infrastructure expansion, and deforestation were the five major proximate causes of LULCC perceived by the households in the study area (Table 6). Both agricultural land expansion and fuelwood extraction were observed as significant proximate causes of LULCC in the study area (Table 6). This finding is in agreement with [28]. Ref. [15] reported that agricultural expansion was the leading LULCC associated with all cases of deforestation, while fuelwood extraction was the second most frequent proximate cause of deforestation (89% of cases), followed by infrastructure expansion (66% of cases) and other factors (31% of cases). As a result, it may cause serious environmental consequences (e.g., land degradation). The local communities of the study used wood extraction for fuelwood, income generation, housing construction, and maxi-

mizing agricultural land. Furthermore, the conversion of forests to pasture and cropland has been reported as the most important proximal cause of tropical deforestation [43]. Therefore, deforestation causes a tremendous negative impact on watershed functions such as reduced peak flows, greater dry season flows, landslide prevention, improved water quality and reduced sedimentation of reservoirs and waterways [15]. At the proximate level, these factors influence and affect the state of cropland expansion, overgrazing, and infrastructure extension [40]. This expansion has been largely achieved at the expense of forests and shrublands [13,23] due to increasing demands for food. Agricultural activity was thus the most important cause of LULCC in terms of severity, followed by fuelwood extraction, road network development, settlement expansion, and bush fire [44]. In this study, fuelwood extraction was the primary cause of LULCC followed by agriculture and infrastructure expansion.

**Table 6.** The proximate causes of LULCC based on logit regression model.

| Causes | Coef. | Std. Err. | Z | *p* > z | [95% Conf. | Interval] |
|---|---|---|---|---|---|---|
| Agricultural land expansion | 1.80 | 1.04 | 1.73 | 0.08 * | −0.24 | 3.85 |
| Fuelwood extraction | 1.52 | 0.61 | 2.51 | 0.01 ** | 0.33 | 2.71 |
| Overgrazing | 0.35 | 0.53 | 0.67 | 0.50 | −0.68 | 1.39 |
| Deforestation | −0.42 | 0.52 | −0.81 | 0.42 | −1.43 | 0.60 |
| Infrastructure expansion | 0.24 | 0.47 | 0.52 | 0.61 | −0.68 | 1.17 |

Note: ** and * represent the statistical significance levels at 5% and 10%, respectively.

More than 80% of the fuelwood was extracted from the forest and shrub/bushlands [45]. However, eucalyptus plantations were one of the possible solutions to use as a fuelwood versus cutting the forests. Eucalyptus plantations fulfill the shortage of fuelwood and construction materials in various parts of Ethiopia [4]. It is difficult to implement the right policies and institutional structures to slow down deforestation in developing countries [46]. However, this study suggested that attitudinal change on the household level toward the short- and long-term negative impacts of deforestation could reduce both the rate and magnitude of deforestation. Two strategies are commonly deployed to control agricultural expansion and promote nature and conservation and benefits [47]. These are: (1) land uses zoning and (2) agricultural intensification. Intensifying agricultural land, in contrast, is thought to spare land from the plow because higher yields decrease the area that needs to be put under agriculture to reach a given production level [36]. The same author added that the implementation or performance of the two aforementioned strategies was generally considered to be under the control of national policies, at least as they are treated in land use change modeling and policy formulations. From 1984 to 2015, the agricultural land of the study area has been intensifying from 6076 to 6232 ha annually at a rate of five hectares. The reduction of shrub/bushland and barren land is one of the contributors to agricultural land intensification in the study area.

### 3.4. Underlying Driving Factors

Underlying driving forces are fundamental social processes, such as human population change or agricultural policies that underpin the proximate causes and either operate at the local level or have an indirect impact from the national or global level due to demographic, economic, technological, policy and institutional and cultural factors [40,41]. Similarly, there are three major underlying driving forces of LULCC such as climate change, population growth, and economic development [48]. However, this study found six prominent underlying factors of LULCC such as population growth, lack of land use policy, climate variability or change, persistent drought, topography and lack of employment. Persistent drought, rapid population growth, and climate variability or changes were the significant and prominent underlying driving factors of LULCC in the study area (Table 7). Drought had 13.5 times higher chance to influence the LULCC as compared to those with-

out drought, similarly population growth had 12 times higher chance of perceiving that LULCC compared with those with no population growth, and climate variability had a 4.5 times higher chance of perceiving that LULCC is due to climate variability as compared to those who did not perceive this. The authors in [30] reported that persistent drought is one of the major driving forces of land use and land cover changes occurring in the Rift Valley dry lands of Ethiopia. This is because the changes in precipitation and temperature and its interactions with various land use and land cover types led to the incidence of drought. Furthermore, the authors of [49] reported that climate change and economic development have a profound influence on LULCC.

**Table 7.** Underlying driving forces of LULCC.

| Factors | Coef. | Std. Err. | Z | *P* > z | [95% Conf. | Interval] |
| --- | --- | --- | --- | --- | --- | --- |
| Persistent drought | 2.61 | 0.77 | 3.38 | 0.00 * | 1.09 | 4.12 |
| Population growth | 2.49 | 1.13 | 2.21 | 0.03 * | 0.28 | 4.69 |
| Lack of land use policy | −0.667 | 0.61 | −1.10 | 0.27 | −1.86 | 0.52 |
| Climate variability/change | 1.50 | 0.60 | 2.50 | 0.01 * | 0.32 | 2.68 |
| Topography | 0.58 | 0.54 | 1.07 | 0.29 | −0.48 | 1.64 |
| Lack of employment | 0.17 | 0.55 | 0.30 | 0.76 | −0.90 | 1.24 |

Note: * represents statistical significance level at 5% ($p < 0.05$).

In the study area, recurrent drought occurs almost once every two to three years. The combined effects of drought, settlement expansion, land tenure policy, and livestock disease cause the LULCC [18]. In addition, farmers destroy and sell trees to produce charcoal and firewood to cope with drought. Due to this reason, the land use and land cover types of the study area have changed from time to time. This is why the smallholder farmers in the study area noted drought as their main drivers of LULCC. Furthermore, the authors of [13,26] reported that climate variability and change are other drivers of LULCC at different temporal and spatial levels. Because they influence land use in multiple ways, for example, rising sea levels, periods of intensified rainfall or drought, changing temperatures and humidity affect the conditions of agricultural production [50]. Topography, lack of land use policy and lack of employment were not statistically significant but were prominent underlying forces of LULCC.

Population growth was another major underlying driving force of LULCC because human activities have significantly influenced the land for maximizing agricultural production and other purposes. The population has proportionally increased. The demands for resources for centuries, resulting in the conversion of natural environmental conditions, have also increased [51]. The reduction in forestland was reported due to the expansion of cultivated land driven by high population growth [33]. Population growth thus drives unsustainable intensification in smallholder agriculture [52]. Both population growth and changing farming practices were noted as the major driving factors of LULCC in Guder and Aba Gerima watersheds, Ethiopia [2,4]. Therefore, when population increases and land scarcity becomes critical, nonfarm activity could therefore be useful to eradicate poverty for land-poor farmers as well as for a primary source of livelihood for the new generation of rural residents [20]. One of the immediate consequences of population growth is the loss of agricultural land [53]. However, in the study area, both the population density and agricultural land (croplands) have increased. Specifically, the population is growing at an alarming rate. Consequently, the area under cropland was also expanding against the shrinking of water bodies, grasslands and shrub/bushlands.

Population density is one of the most important underlying driving factors of LULCC in the clearing of forests for the expansion of agricultural land [13,54] and built up areas. On a global scale, deforestation has been linked to increases in population density and per capita consumption [10]. This may significantly affect the coverage of forests and grassland resulting from the reduction and abandonment of fallow systems [55] and the overall climate condition of the area. Studies have indicated that the population density of Ethiopia

during the period of 1994 and 2007 was about 47.3 and 65.3 persons per km$^2$, and this value increased to 99.0 persons per km$^2$ in 2015. The changes in population densities during the second half of the 20th century clearly had an effect on land use and land cover, resulting in shrinking forests and grassland, expansion of cultivated areas, and intensified use resulting from reduction and almost complete abandonment of fallow systems [56]. Figure 3 shows that population density is increasing in the study area. In the years 1994(5), 2007, and 2015, the overall population density of the study area was about 65.3, 82.6, and 93.0 persons per km$^2$, respectively. For example, the population density of the lowland area such as Yalo, Megale and Gulina during the period 1994 was less than 14.5 persons per km$^2$; however, in 2015, the figure rose to 50.5 persons per km$^2$ (Figure 3). Similarly, in the midland area, the 1995 population density was about 49.2; however, in 2015, the figure increased to 133.9 persons per km$^2$. Furthermore, in the highlands area, the minimum density in 1995 was about 57.4; however, the value increased to 146.6 persons per km$^2$ during the period 2015. In the study area, the population density was higher in the highlands area than in the lowlands and midlands. Therefore, an increase in population density may escalate the demand for land, as it is a major driving force of the LULCC in Ethiopia [55].

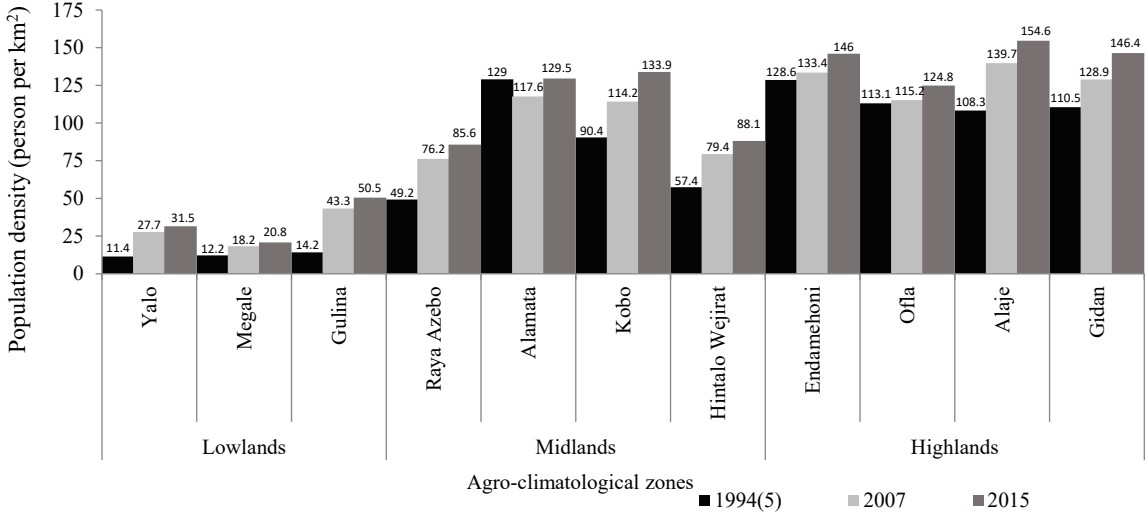

**Figure 3.** Population density of the study area.

In addition, Ref. [40] stated that policies could affect land use directly. On the other hand, the absence of policy may also cause improper practices in land use as well as land management by the smallholder farmers in the country in general and the study area in particular.

## 4. Conclusions

The dynamics of LULCC have been a serious environmental challenge in the study area. Our findings indicated that agricultural land expansion, fuelwood extraction, deforestation, overgrazing and development of infrastructure were the proximate causes of LULCC in Raya Valley. The increase in agricultural land at the expense of forest and other vegetation cover may lead to serious land degradation problems in the study area. On the other hand, drought, climate change, population growth, lack of land use policy, topography, and unemployment were the underlying driving factors. The potential determinant factors that affect the land cover change were fuelwood extraction and agricultural land expansion. Persistent drought, population growth and climate variability were investigated as the determinant factors of LULCC in the study area. The study also reported that there was a significant association among the various determinant factors of LULCC. Therefore, this study suggested that a detail investigation on the implication of LULCC should be carried out based on the identified proximate causes and underlying driving forces in Raya Valley

for designing and implementing better land use planning, land management strategies and policy interventions.

**Author Contributions:** Conceptualization, E.G.; methodology, E.G. and S.M.; software, S.M.; validation, O.D., R.S. and E.G.; formal analysis, E.G.; investigation, E.G.; resources, E.G.; data curation, E.G.; writing—original draft preparation, E.G.; writing—review and editing, E.B., A.Z., P.M., O.D., R.S. and E.G.; visualization, S.M.; supervision, O.D., R.S., E.G. and E.S.; project administration, E.G.; funding acquisition, E.G. All authors have read and agreed to the published version of the manuscript.

**Funding:** Mekelle University, TreccAfrica II, and Open Society Foundation-Africa Climate Change Adaptation Initiative (OSF-ACCAI, grant number: OR2016-30576).

**Institutional Review Board Statement:** Not applicable.

**Data Availability Statement:** The data presented in this study are available on request from the corresponding author.

**Acknowledgments:** We are indebted to the support of Mekelle University, TreccAfrica II, OSF–ACCAI, and CESET project.

**Conflicts of Interest:** The authors have declared that there are no conflict of interests.

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
