# Peer review of "Land Use and Land Cover Change Determinants in Raya Valley, Tigray, Northern Ethiopian Highlands"

_agriculture, doi:10.3390/agriculture13020507_

Round 1

Reviewer 1 Report

The selected research topic is quite interesting, and it has also been done in an appropriate way. However, some points required to address in order to enhance the quality of the study. 

The line number is not appearing in the manuscript and it make some obstacles to giving line-by-line comments.

The information that should be included in the result and discussion section of the study seems to be included in the study methodology. Please check section 2.4 “An increased annual change rate trend was observed in shrub/bush lands by 150.3 km2 (5.33%), grasslands 21.6 km2 (6.59%), built-up area 8.8 km2 (9.19%), forestland 6.6 km2 (5.73%), barren land 2.5 km2 (0.15%), and water body 0.6 km2 (1.19%), respectively. However, the croplands and…..”. I believed that figure 2 and table 1 should on  results and discussion section.

There is nothing new in this work. There have been a thousand research published in this field, each with different sophisticated methods and goals. This study will not add anything to the field, just shift the focus and make use of preexisting algorithms.

As a whole, the quality of the writing is low, and it deviates too far from accepted academic standards to be considered acceptable (extensive English editing is required). This version needs considerable work since there are issues such as redundant language and improper and inconsistent verb tenses. 

Wish you all the best 

Author Response

Response to Reviewer 1 Comments

Point 1: The selected research topic is quite interesting, and it has also been done in an appropriate way. However, some points required to address in order to enhance the quality of the study. 

Response 1: Dear reviewer, thank you for your feedback. We addressed your constructive comments to enhance the quality of the manuscript.

Point 2: The line number is not appearing in the manuscript and it make some obstacles to giving line-by-line comments.

Response 2: Dear reviewer, we added line numbers in each pages to avoid some challenges.

Point 3: The information that should be included in the result and discussion section of the study seems to be included in the study methodology. Please check section 2.4 “An increased annual change rate trend was observed in shrub/bush lands by 150.3 km2 (5.33%), grasslands 21.6 km2 (6.59%), built-up area 8.8 km2 (9.19%), forestland 6.6 km2 (5.73%), barren land 2.5 km2 (0.15%), and water body 0.6 km2 (1.19%), respectively. However, the croplands and…..”. I believed that figure 2 and table 1 should on  results and discussion section.

Response 3: Dear reviewer, the figure 2 and table 1 are considered as input (data) not result of our manuscript. However, Gidey et al. (2017) have published the results. Now, we used the resuls of land use and land cover change to investigate why it is chnging from time to time. In this case, the results of Gidey et al. (2017) were used as row data and based on that we conducted a research and prepared the current manuscript.

Point 4: There is nothing new in this work. There have been a thousand research published in this field, each with different sophisticated methods and goals. This study will not add anything to the field, just shift the focus and make use of preexisting algorithms.

Response 4: Dear reviewer, this manuscript provides good insights to comprehend the majior drivers of land use/cover change in the study area. The study area has been seriously affected by the changes of land use/cover and there are no systematically conducted studies that address the driving forces in the area. As a result, the land administrators have conventionally managing the lands. Therefore, this study will support th community to comprehend the fact why is their land use/cover changing and its potential implication. As you have also indictaed, there are studies worldwide that discussess about the driving forces because it is must to see always when ever you research the histrorical/future land use/cover changes. This can help to adjust land policy as well as land administration systems. The article models the drivers of land cover change in Ethiopia using community survey data. We identified the major proximate causes of land use land cover change as identified by survey responses. The data set can reveal essential information regarding communities’ perspective on the causes of land conversion. This newfound knowledge can be then used to guide land use policies and help improve existing landmanagement strategies.

Point 5: As a whole, the quality of the writing is low, and it deviates too far from accepted academic standards to be considered acceptable (extensive English editing is required). This version needs considerable work since there are issues such as redundant language and improper and inconsistent verb tenses. 

Response 5: Dear reviewer, we improved the quality of our manuscript as per your suggestion. The manuscript is checked and improved by a professional english writer and professor from our university. Hopefully this will signficantly improve the writing of the manuscript.

Point 6: Wish you all the best 

Response 6: Dear reviewer, thank you for your good wishes and we have learned a good lesson from your constructive comments. Thank you.

Best regards,

Authors of this manuscript.

Reviewer 2 Report

Please see document attached. 

Author Response

Response to Reviewer 2 Comments

Point 1: The article titled “Land use and Land cover change determinants in Raya valley–Tigray,Ethiopia” models the drivers of land cover change in Ethiopia using community survey data. The authors identify the major proximate causes of land use land cover change as identified by survey responses. The topic is important, and the data set can reveal essential information regarding communities’ perspective on the causes of land conversion. This newfound knowledge can be then used to guide land use policies and help improve existing landmanagement strategies. However, the study would benefit from a major review before it can be considered for publication.

Response 1: Dear reviewer, thank you for your feedback. You have clearly indicated the importance of this manuscript based on our findings. All your constructive feedbacks are considered to enhane the quality of our mansucript. Thank you again.

Point 2: The introduction and conclusion sections could be significantly improved. In the introduction important studies are not referenced, and the conclusion simply restates information that was already stated rather than providing an in-depth analysis of the study findings in a larger context of land management strategies and policy interventions.

Response 2: Dear reviewer, we improved the introduction and conclusion sections of our manuscript as per your suggestion. Important references are also added in the introduction and other sections of the manuscript.

Point 3: Some of the figures could also be improved, specifically Figure 2.

Response 3: Dear reviewer, we tried to improve the figure by adding some missing information, e.g., North arrow.

Point 4: The overall flow of the manuscript can be improved, and a more in-depth discussion of the figures, and tables can help the reader better understand survey responses. Adding a few examples of the survey questions could also improve the manuscript..

Response 4: Dear reviewer, it is correted as per the suggestion. Some of the survey questions that has been asked to the respondesnts were. 1) Has there been any change in your land use system? What ara the majior LULCC driving forces? Why it is changing from time to time?

Point 5: Extensive editing is needed in to correct recurrent issues such as LULC when in fact it should be LULCC – land use land cover change. 

Response 5: Dear reviewer, we corrected the issues according to your suggestions.

Point 6: Please add line numbers to the draft manuscript so that reviewers can refer to specific line numbers when providing comments

Response 6: Dear reviewer, we added line numbers for each statements (sections).

Point 7: In the abstract please fix the error “LULC The determinants” – here the period is missing, and it should be LULC change not simply LULC. Also consider changing LULC change to LULCC –land use land cover change

Response 7: Dear reviewer, we addedd the missing period (.) and considered LULCC –land use land cover change throught the manuscript.

Point 8: In the introduction fix the way percent is spelled out “per cent”. This needs to be fixed throughout the entire manuscript.

Response 8: Dear reviewer, it is corrected accordingly throughout the manuscript.

Point 9:  Consider changing LULC change to LULCC

Response 9: Dear reviewer, we changed as per your feedback.

Point 10: [6] Suggested is a very strange way to start a sentence – cite the actual name of the lead author of the study or consider rephasing the sentence.

Response 10: Dear reviewer, it is corrected now.

Point 11: The paragraph that starts with: “In Africa, few studies on LULC driving forces have been
conducted…” if the reader spells out the acronym LULC then it reads as “In Africa, few studies
on land use land cover driving forces have been conducted” – the word change is missing.

Response 11: Dear reviewer, it is improved as per your constructive feedback.

Point 12: [16] – cite the study rather than just the number so that the sentence does not start with a number. Since the authors are referring to a specific study it would be helpful for the reader to see the name of the study rather than just seeing a number and having to go to the reference list to see the details of the study that is described in the text.

Response 12: Dear reviewer, you are correct, but we followed stricktly the style of the journal.

Point 13: After the sentence – “The observed LULC change was driven mainly by the combination of both proximate cause and underlying driving factors.” Consider defining proximate causes and
underlying driving factors and mention the ways in which one is different from the other.

Response 13: Dear reviewer, we defined the proximate causes and underlying driving forces of LULCC. We cited proporly the reference that we have used to define them.

Point 14: In the sentence - “has been significantly affected by the cultivation and productivity of the land.” The word “by” is missing.

Response 14: Dear reviewer, it is corrected now. Thank you.

Point 15: “Ethiopia For instance” – the period here is missing again.

Response 15: Dear reviewer, it is corrected now. Thank you.

Point 16: “the driving forces of LULC” needs to be - the driving forces of LULC change if the LULC acronym is not changed to LULCC. Again, the word change is missing from “characterize the perceptions of local people in LULC driving forces.”

Response 16: Dear reviewer, they are corrected as per the suggestion.

Point 17: “In addition, a detailed understanding of the drivers of land use and land cover change and their inter–linkages improves intervention and avoid a decline in natural resources.” This sentence needs to be edited. Change “avoid” to “avoids” since this is referring to the detailed
understanding.

Response 17: Dear reviewer, it is correted as per your feedback.

Point 18: In addition, keep the reference to LULC consistent throughout the manuscript
either use the acronym or spell it out.

Response 18: Dear reviewer, it is corrected as per your suggestion.

Point 19: Change “Rainfall was erratic and bimodal” to present tense – Rainfall is. I suggest changing all the tenses of the Study Site description from past tense to present tense.

Response 19: Dear reviewer, it is corrected.

Point 20: Consider rephrasing the sentence that starts with “In this area, population growth has been reported as one of the main determinants of land cover change and causes a greater loss in the majority of LULCs”. The acronym does not read as intended in this case. “Agriculture” does not need to be capitalized.

Response 20: Dear reviewer, it is corrected.

Point 21:  For Figure 1 what does m.a.s.l mean? In addition to this, for the elevation gradient, can you reclassify the classes so that the numbers are round numbers instead of 324, 1275… etc. so maybe have 350, 1500, or something easier for the reader to conceptualize. For the insert on the upper right side please make the “Ethiopia” label easier to visualize. If possible, consider adding the locations of the communities where the survey was conducted. Consider changing the color of the roads to black or another color that would be easier to distinguish on top of the elevation color gradient.

Response 21: Dear reviewer, m.a.s.l stands for meters above sea level. We reclassified the elevation of the study area according to Ethiopian Ministry of Agriculture (MoA) agro–climatological zones characterization. We also improved the lable of the country to be better visualized in the map. We tried to change the color of the roads to black, but it was matching with the boundary color of each districts within the study area. Thereforee, we prefered to remain as it is. The survey was conducted in each agro–climatological zones of the study area (please see Table 1 for a detail). This table is linked with Figure 1.

Point 22:  A basemap added to Figure 1 could also help the reader better understand the geographic and spatial characteristics of the area

Response 22: Dear reviewer, we added the rgional administrative boundary and topography as a basemap to comprehend the spatial charactetics of the study area (please see Figure 1).

Point 23:  It is rather strange to start a sentence with “[27] sampling” please rephrase. Similar comment for all the other sentences that start with […]. The sentence that says: “And why is changing from time to time?” to “why it is changing from time to time?

Response 23: Dear reviewer, it is corrected accordingly.

Point 24:  “LULC dynamics” change to “LULC change dynamics” Please provide a higher resolution version of Figure 2. Is Figure 2 adapted from a different study? If so, please make that clear in the figure caption.

Response 24: Dear reviewer, Figure 2 was taken from other study which was published in 2017 by Gidey et al. (2017). However, we tried to include some of the requested feedbacks.

Point 25:  In Figure 2 what do the acronyms in the legend mean? If the acronyms correspond to land cover classes, this needs to be clearly stated in the figure legend or the acronyms need to be described in the figure caption.

Response 25: Dear reviewer, the acronyms in the legend stands for Cropland (Cl):, Forest land (Fl), Shrub/bushland (Shl/bl), Built–up area (Bu), water body (Wb), Grassland (Gl), Barren land (Bl), Floodplain (deposition) area (Fl:). Yes, the land cover classes were taken from other study which was published in 2017 by Gidey et al. (2017).

Point 26:  The acronyms are displayed in the Note in Table 2, but the description needs to be associated with the Figure. The scale bar in Figure 2 should be added to the maps, and the north arrow is missing. It is not clear what (c) is referring to since it is not described in the figure legend.

Response 26: Dear reviewer, the description of each land cover types are included as a separate table in Table 3. The scale bar in Figure 2 was added to the maps, but the north arrow was missing. Now, we added the north arrow in fingure 2.

Point 27:  “Such as chi–square” S does not need to be capitalized.

Please fix the editing of Equation 2.

Response 27: Dear reviewer, it is correted.

Point 28:  “land use and land cover change (LULC)” the acronym is not correct, it should be LULCC. Please rephase “female household heads cover only 42”

Response 28: Dear reviewer, The acronym of land use and land cover change is corrected to LULCC throughout the manuscript. We also represhed the “female household heads cover only 42” statement to “…..while the female headed households were 42”.

Point 29: Section 2.5 - correct “codded”. Please describe the logit model and consider citing relevant studies that used a similar approach 

Response 29: Dear reviewer, it is correcteed. 

Point 30:  Section 3.2 – add change after “driving forces of LULC”. “Fuelwood” does not need to be capitalized.

Response 30: Dear reviewer, it is corrected accordingly.

Point 31: Table 5: Please explain how Topography is driving LULC change. Topography is static (not changing) as compared to the other variables in Table 5. Since this is not a dynamic parameter, how is this a driving force of LULCC (land use land cover change)?  

Response 31: Dear reviewer, as you known that, LULCC caused by the effects of natural and cultural factors. For instance, climate, soil, and topography hava been considered as natural factor, while population growth, economic growth, and policies are cultural factotrs. Therefore, in our manuscript topography identifid as one of the underlying driving fcators of LULCC.

Point 32: In the sentence: “Population growth was another major underlying driving force of LULC.” The word change is missing.

Response 32: Dear reviewer, We added the missing word.

Point 33: Consider editing the Conclusion section of the study to say more than a summary of the findings. Rather, try and write the Conclusion section by linking the findings of the study within a larger general framework and provide potential implications that would result from these findings.

Response 33. Dear reviewer, We improved the conculusion section of the study as per the given feedback.

Point 34: How can the results of this work be used to guide land management strategies, and/or policy? What do the results of the study highlight in the context of existing literature. As it is currently written the conclusion simply repeats the information presented in the abstract and does not synthesize the results or provides any other concluding information. Potentially consider adding information on next steps or study limitations, data needs, and management implications. What other studies have identified similar proximate causes of land use land cover changes?

Response 34. Dear reviewer, the ultimate objective of this study is just to scrutinize the proximate causes and underlying driving forces of LULCC scientifically. Based on this finding, land use planners and  land managers could develop and implement approporate strategis that can diminish the impact of LULCC.

Point 35: In the methods section consider adding more details regarding the study design. Add information of the demographics of the community/communities surveyed. For example, what is the ration of male to female in the communities’ survey? Consider adding information that would help the reader understand whether or not the survey responses represent a representative sample of the communities studied. Why is the youngest age of survey respondents 41?

Response 35: Add information of the demographics of the community/communities surveyed. Dear reviewer, the general demographic haracterstics of the study area is presented in section 2.1 Table 1. Table 2 is also clearly indicating the sampled households surveyed. Why is the youngest age of survey respondents 41? Because our study was focusing on older age group to provide us detail information about the reason why it is changing the land use and land cover types of the study area. That is why our youngest age of survey respondent was 41.

Point 36: The Introduction section would also benefit from further editing. The introduction does not cite some studies that are very relevant in the field of land system science. Numerous studies discuss the proximate cause and underlying driving factors of LULCC. Consider reviewing the flowing:

Geist, H. J., & Lambin, E. F. (2002). Proximate causes and underlying driving forces of tropical
deforestation. BioScience, 52(2), 143–150.by
Meyfroidt, P. (2016). Approaches and terminology for causal analysis in land systems science.
J. Land Use Science. 11 (5), 501–522.
Turner, B. L., Lambin, E. F., & Reenberg, A. (2007). The emergence if land change science for
global environmental change and sustainability. PNAS, 105(7):20666-20671

Geist, H. J., & Lambin, E. F. (2002). Proximate Causes and Underlying Driving Forces of
Tropical Deforestation. BioScience, 52(2), 143. Also relevant to the study area:
Betru, T., Tolera, M., Sahle, K., & Kassa, H. (2019). Trends and drivers of land use/land
cover change in Western Ethiopia. Applied Geography, 104, 83–93.
https://doi.org/10.1016/j.apgeog.2019.02.007

Response 36: Dear reviewer, we considered the aformentioned literatures. Yes, they are helpful. Of cours, some of the manuscripts were already reviewed in our manuscript.

Thank you for your critical review and invaluable feedback. We have learned a lot from the comments.

Best regards,

Authors of this manuscript.

Round 2

Reviewer 1 Report

The manuscript has been improved. I have no comments 

Author Response

There is no feedback in round 2 from reviewer 1.

Reviewer 2 Report

Please see file attached. 

Author Response

Thank you for the comments. The comments raised are dealt line by line. The detail reply to the comments are attached.
